# Seaweed Proteins: A Step towards Sustainability?

**DOI:** 10.3390/nu16081123

**Published:** 2024-04-10

**Authors:** Leonel Pereira, João Cotas, Ana Marta Gonçalves

**Affiliations:** 1Marine Resources, Conservation and Technology, Marine Algae Laboratory, Centre for Functional Ecology—Science for People & the Planet (CFE), Department of Life Sciences, University of Coimbra, 3000-456 Coimbra, Portugal; jcotas@uc.pt (J.C.); amgoncalves@uc.pt (A.M.G.); 2Department of Biology and CESAM—Centro de Estudos do Ambiente e do Mar, University of Aveiro, 3810-193 Aveiro, Portugal

**Keywords:** macroalgae proteins, alternative protein sources, nutritional composition, extraction methods, functional properties, sustainability, environmental advantages

## Abstract

This review delves into the burgeoning field of seaweed proteins as promising alternative sources of protein. With global demand escalating and concerns over traditional protein sources’ sustainability and ethics, seaweed emerges as a viable solution, offering a high protein content and minimal environmental impacts. Exploring the nutritional composition, extraction methods, functional properties, and potential health benefits of seaweed proteins, this review provides a comprehensive understanding. Seaweed contains essential amino acids, vitamins, minerals, and antioxidants. Its protein content ranges from 11% to 32% of dry weight, making it valuable for diverse dietary preferences, including vegetarian and vegan diets. Furthermore, this review underscores the sustainability and environmental advantages of seaweed protein production compared to traditional sources. Seaweed cultivation requires minimal resources, mitigating environmental issues like ocean acidification. As the review delves into specific seaweed types, extraction methodologies, and functional properties, it highlights the versatility of seaweed proteins in various food products, including plant-based meats, dairy alternatives, and nutritional supplements. Additionally, it discusses the potential health benefits associated with seaweed proteins, such as their unique amino acid profile and bioactive compounds. Overall, this review aims to provide insights into seaweed proteins’ potential applications and their role in addressing global protein needs sustainably.

## 1. Introduction

The global demand for protein continues to rise with increasing populations and changing dietary preferences. Traditional sources of protein, such as meat, poultry, and dairy, face challenges related to environmental sustainability, resource depletion, and ethical concerns [1]. As a result, there is growing interest in exploring alternative sources of protein that are both nutritious and sustainable. Seaweed, a diverse group of marine algae, presents a promising solution due to its high protein content and minimal environmental impact [2].

Seaweed as a high-protein food refers to various marine algae that contain a significant amount of protein relative to their overall composition. Seaweed is known for its rich nutritional profile, which includes vitamins, minerals, antioxidants, and other beneficial compounds [3]. While the protein content can vary depending on the type of seaweed, it generally ranges from 10% to 30% of its dry weight. Seaweed is considered a valuable source of plant-based protein, especially for individuals following vegetarian or vegan diets, as it provides essential amino acids necessary for human health. Incorporating seaweed into the diet can contribute to meeting daily protein requirements while also offering other health benefits associated with its nutrient content [4].

With burgeoning populations, environmental concerns, and shifts in dietary preferences, traditional protein sources such as meat and dairy are being scrutinized for their sustainability and ethical implications [3]. In response, researchers and food innovators are exploring diverse avenues to meet protein needs while mitigating the environmental impact of food production. One promising avenue lies in harnessing the potential of seaweed proteins, which offer a sustainable and nutritious alternative to conventional protein sources [4].

Traditionally valued for its culinary uses in various Asian cuisines, seaweed is now garnering attention for its protein content and sustainability credentials [5]. Seaweed cultivation requires minimal freshwater, land, and fertilizers, making it a highly sustainable protein source compared to land-based agriculture [6]. Moreover, seaweed cultivation can sequester carbon dioxide and mitigate ocean acidification, further enhancing its environmental benefits [7].

In recent years, the exploration of seaweed proteins as a viable alternative to animal-derived proteins has gained momentum [8]. Seaweed proteins boast a unique amino acid profile, containing all the essential amino acids necessary for human nutrition [9]. Additionally, seaweed proteins are rich in vitamins, minerals, and bioactive compounds, offering a holistic nutritional package [10]. With advancements in processing technologies, seaweed proteins can be extracted and incorporated into a variety of food products, including plant-based meats, dairy alternatives, and nutritional supplements. This versatility makes seaweed proteins an attractive option for addressing the protein needs of diverse consumer demographics, including vegetarians, vegans, and individuals seeking sustainable dietary choices [11].

Furthermore, the sustainable cultivation of seaweed holds promise for addressing key challenges in global food security and environmental sustainability. Seaweed cultivation requires no arable land and minimal inputs, making it accessible to coastal communities and land-scarce regions [12]. By leveraging seaweed cultivation as a means of protein production, nations can reduce pressure on terrestrial ecosystems, preserve biodiversity, and promote resilience in the face of climate change. Moreover, the cultivation of seaweed can create employment opportunities and support livelihoods in coastal communities, contributing to economic development and poverty alleviation [13].

The growing demand for alternative protein sources underscores the need for sustainable and innovative solutions to meet the nutritional needs of a burgeoning global population [14]. Seaweed proteins offer a compelling pathway towards achieving this goal, combining nutritional excellence with environmental stewardship [15]. As research and development efforts continue to unlock the full potential of seaweed proteins, they hold the promise of revolutionizing the food industry and fostering a more sustainable and equitable future for generations to come [16,17,18,19].

This review aims to explore the seaweed potential towards a more sustainable human welfare future, reducing the pressure from traditional protein sources, with the possibility of reducing human health problems.

## 2. Nutritional Composition of Seaweed

The chemical composition of macroalgae presents great variability as it can be influenced by the species, stage of development, geographical location, habitat, season, and nutrient content in the growth medium, among other environmental factors. Additionally, different sampling and drying methodologies can affect the biochemical composition and, consequently, the nutritional value [20]. Algae are a good source of micronutrients such as vitamins (e.g., vitamin A, B_1_, B_2_, B_3_, B_6_, B_12_, C, D, E, pantothenic acid, and folic acid), sterols, and minerals (e.g., calcium, magnesium, potassium, iodine, sodium, phosphorus, nickel, chromium, selenium, iron, zinc, manganese, copper, lead, cadmium, mercury, and arsenic) [21]. Magnesium, nickel, chromium, iron, zinc, manganese, and copper are essential nutrients for various physiological and biochemical functions of the body [22]. However, they should be consumed in adequate quantities, as excessive exposure can cause acute or chronic toxicity [23]. The remaining minerals (lead, cadmium, mercury, and arsenic) are considered undesirable [24]. This mineral richness is related to the algae’s ability to retain inorganic matter, which represents up to 36% of dry weight in some species [25].

Algae are one of the few non-animal sources of vitamin B_12_. For example, it is reported that *Porphyra* spp. (Figure 1) contains 32.26 to 133.8 µg per 100 g of dry weight, which equates to 1.61 µg (64% RDA) to 6.69 µg (268% RDA) in a 5 g portion of *Porphyra* spp. [25], considering that the recommended daily dose (RDA) of vitamin B_12_ is 2.5 µg for an adult, according to Regulation (EU) No 1169/2011. Thus, algae containing vitamin B_12_ can be useful for individuals following a vegan diet [26].

In addition to micronutrients, seaweed is a source of macronutrients, including proteins and amino acids (essential and non-essential), carbohydrates (mostly fiber), and fat (with special interest in polyunsaturated fatty acids) [16]. Overall, red algae contain high levels of protein, up to 47% (*w*/*w*) on a dry weight basis. Green algae contain moderate amounts, ranging from 9 to 26% (*w*/*w*) on a dry weight basis, while brown algae contain a much lower protein content, ranging from 3 to 15% (*w*/*w*) on a dry weight basis [27]. According to the literature, during the winter and early spring, macroalgae have the highest protein content [28]. Cultivated algae have a higher protein content compared to wild algae because they grow in environments that are often nutrient-limited, while cultivated algae grow in nutrient-rich water from terrestrial systems [29].

It should be noted that the protein content in algae is often estimated using a conversion factor of 6.25 (the Kjeldahl method), based on the assumption that most of the nitrogen found in the sample occurs as protein nitrogen [30]. However, this conversion factor may overestimate the protein content due to the presence of variable amounts of non-protein nitrogen in the sample (e.g., chlorophyll, nucleic acids, free amino acids, and inorganic nitrogen). In more recent studies, other conversion factors have been proposed that vary with species and season [31]. Angell et al. (2016) propose a universal conversion factor of 5 a rounded value of the global average of conversion factors (4.97), based on the ratio of total amino acids to total nitrogen. This study included the analysis of 103 species, covering three taxonomic groups, various geographic regions, and different physiological states [32]. Lourenço et al. [33] calculated the nitrogen-to-protein conversion factor for the three different taxonomic groups in the same way and obtained the following results: 5.13 (green algae), 5.38 (brown algae), and 4.59 (red algae). The values suggest that red algae contain a higher amount of non-protein nitrogen [33]. However, both studies show that the widespread use of the conversion factor 6.25 is inadequate for seaweeds.

Amino acid composition is essential for determining the value of proteins in the human diet, particularly to achieve adequate intake of essential amino acids [34]. Algal proteins contain significant amounts of essential amino acids, representing almost half of the total amino acids [25]. However, tryptophan and lysine are generally limiting amino acids in most algal species [8]. Additionally, cysteine is usually low in many species of seaweed and is generally undetectable [25]. In *Ulva rigida* (Figure 2), leucine, phenylalanine, and valine are the major essential amino acids, and histidine levels are similar to those found in legumes and eggs [35,36]. There is a notable similarity in the total amino acid composition of the *Ulva* genus with egg ovalbumin [37]. In most analyses of amino acid composition in seaweed, glutamic acid and aspartic acid are the major amino acids and also contribute to their characteristic “umami” flavor [38,39].

The fat content of macroalgae tends to be low relative to the total dry weight. The percentage of fat content is higher in the winter and lower in the summer, and the fatty acid composition varies with the season [40]. It is noteworthy that *Porphyra* spp. has the lowest content of saturated fatty acids (17.4% of total fatty acids) [41].

The carbohydrate content tends to be relatively high; however, macroalgae cannot be considered an energy food because the digestibility of these carbohydrates is low. Typical polysaccharides of red algae include floridean starch, cellulose, xylan, mannan, and sulfated galactans (carrageenans and agars) [42]. In the *Porphyra* genus, porphyran is the predominantly sulfated polysaccharide found. Most of these polysaccharides are not digestible by the human gastrointestinal tract and are therefore considered dietary fiber [43]. In *Gracilaria* spp. (Figure 3), for instance, the dietary fiber content varies from 23.5% to 64% in dry weight, values that surpass the fiber content in most fruits and vegetables [44]. Given the high variability in chemical composition, Table 1 presents the nutritional composition of selected edible seaweeds.

Seaweed cultivation practices significantly influence its nutritional composition, particularly concerning protein content. Cultivated seaweed tends to exhibit higher protein levels compared to wild algae [6,12]. This disparity arises due to the controlled environments of seaweed farms, where nutrient-rich conditions promote robust growth and protein accumulation. In contrast, wild algae often grow in nutrient-limited marine habitats, resulting in a comparatively lower protein content [6].

Various factors during seaweed cultivation contribute to the observed differences in protein content. For instance, the availability of essential nutrients such as nitrogen and phosphorus directly impacts seaweed growth and protein synthesis. Cultivation methods, such as integrated multi-trophic aquaculture (IMTA), where seaweed is grown alongside other marine organisms like shellfish, can enhance nutrient availability and thus increase protein content [13,29].

Additionally, environmental conditions, including temperature, light intensity, and salinity, play crucial roles in shaping seaweed composition. Optimal conditions promote efficient photosynthesis and metabolic processes, leading to higher protein accumulation. Moreover, the developmental stage of seaweed during cultivation can influence protein content, with certain growth phases favoring protein synthesis [6,12].

Understanding these relationships between cultivation methods and seaweed composition is essential for optimizing protein yield and nutritional quality. By elucidating the impact of cultivation practices on seaweed protein content, this review provides valuable insights into harnessing seaweed as a sustainable and nutritious protein source for various applications [28].

## 3. Physiological Importance of Amino Acids

Amino acids are defined as organic substances that contain at least one amine group and one carboxylic group. According to their side chains, amino acids have different biochemical properties and functions [69]. Apart from glycine (the simplest amino acid in nature), all amino acids possess at least one asymmetric carbon and exhibit optical activity [70]. Amino acids can be designated as D- or L-depending on the absolute configuration of the substituents around the asymmetric carbon. L-amino acids are the most frequently found physiological isomers in nature. However, D-amino acids also exist in animals, microorganisms, and plants [71].

There are more than 700 amino acids in nature, but only 20 (α-amino acids) are used as building blocks for proteins. Amino acids used as substrates for polypeptide biosynthesis are called proteinogenic amino acids (e.g., methionine and proline), while amino acids that are not protein building blocks are known as non-proteinogenic amino acids (e.g., citrulline, homocysteine, and hydroxyproline) [72]. However, not all amino acids present in polypeptides can be considered proteinogenic amino acids because modifications may occur after translation, leading to the formation of new amino acid residues. An example is hydroxyproline, which is produced from proline by peptidyl proline hydroxylase after protein synthesis. Amino acids can be classified as nutritionally essential (indispensable) or non-essential (dispensable) [73].

Essential amino acids are those that cannot be synthesized by the body or are synthesized in insufficient quantities to meet the needs and, therefore, must be provided by the diet. Among the 20 proteinogenic amino acids, 9 are considered essential: isoleucine, leucine, valine, lysine, methionine, phenylalanine, threonine, tryptophan, and histidine [71].

Non-essential amino acids are those that can be synthesized by the body in adequate quantities to meet its needs [74]. Amino acids have various physiological functions, including regulation of food intake, gene expression, protein phosphorylation, and cell-to-cell communication. Additionally, amino acids are essential precursors for the synthesis of hormones and low-molecular-weight nitrogenous substances, each with significant biological importance. Thus, a balance of amino acids in the diet and circulation is necessary for organismal homeostasis [35].

### What Are Essential Amino Acids for?

Each of the essential amino acids has specific and vital functions for the proper functioning of the body. They are necessary for tissue building, muscles, some hormones, and enzymes. They are acquired through food or supplementation [75]. Returning to the analogy used to explain the relationship of amino acids (bricks) to proteins (walls), amino acids are more easily absorbed by the body than proteins. This is because proteins are larger and more complex structures, while amino acids consist of a simpler structure [76,77].

## 4. Algae Proteins and Amino Acids

### 4.1. Bioactive Proteins and Amino Acids Derived from Algae

Algae proteins offer a fascinating nutritional composition, characterized by unique amino acid profiles and an array of bioactive compounds [78]. These marine algae, known for their rich biodiversity, have garnered increasing attention for their potential as sustainable and nutritious food sources [21]. One notable aspect of seaweed proteins is their amino acid composition. While the exact profile varies among different species, seaweeds typically contain a wide range of essential and non-essential amino acids [79]. Essential amino acids are those that the human body cannot synthesize on its own and must be obtained through diet. Seaweeds often provide a balanced mix of these essential amino acids, making them valuable components of a balanced diet [80].

Moreover, seaweed proteins are known for their high digestibility and bioavailability. Compared to proteins from terrestrial sources, such as meat or legumes, seaweed proteins are often easier for the body to break down and absorb. This digestibility makes them particularly suitable for individuals with digestive issues or those seeking alternative protein sources [81].

In addition to amino acids, algae proteins contain a variety of bioactive compounds with potential health benefits. These compounds include polysaccharides, peptides, polyphenols, vitamins, and minerals [82]. Some of these bioactive compounds exhibit antioxidant, anti-inflammatory, antimicrobial, and antiviral properties, contributing to the overall health-promoting effects of seaweed consumption [83]. One noteworthy bioactive compound found in seaweed proteins is phycocyanin, a pigment responsible for the blue-green color of certain seaweeds. Phycocyanin has been studied for its antioxidant and anti-inflammatory properties, which may have implications for various aspects of human health, including cardiovascular health and immune function [84]. *Arthrospira platensis* (Cyanobacteria) is a well-known blue-green algae that contains high levels of phycocyanin. In addition to its antioxidant and anti-inflammatory properties, spirulina has been studied for its potential to improve lipid profiles, reduce blood pressure, and enhance immune function [85]. *Aphanizomenon flos-aquae* is a blue-green algae found in freshwater lakes, and like spirulina, this species contains significant amounts of phycocyanin, which contributes to its antioxidant and anti-inflammatory properties. This species has been studied for its potential to support cognitive function, improve mood, and enhance overall well-being [86]. *Nostoc commune* (Cyanobacteria), commonly found in terrestrial habitats, including soil and rocks, produces phycocyanin along with other bioactive compounds. Studies suggest that *N. commune* extracts may have antioxidant, antimicrobial, and anticancer properties, although further research is needed to elucidate their full therapeutic potential [87]. These examples illustrate how algae species containing phycocyanin offer a range of health benefits, including antioxidant and anti-inflammatory effects, which may support cardiovascular health, immune function, and overall well-being [85].

Furthermore, seaweeds are often low in fat and calories, making them an attractive option for individuals seeking to manage their weight or reduce their intake of saturated fats [80]. Incorporating seaweed proteins into the diet can offer a nutrient-dense alternative to traditional protein sources, contributing to a well-rounded and healthy eating pattern [88]. The nutritional composition of algae proteins, including their amino acid profiles and bioactive compounds, underscores their potential as valuable additions to the human diet [89].

As research in this field continues to expand, further exploration of the health benefits and culinary applications of seaweed proteins promises to enhance our understanding of their role in promoting overall health and well-being. For example, Irish moss (*Chondrus crispus*) (Figure 4), a red alga commonly found in the Atlantic Ocean, contains a moderate amount of protein, along with carrageenan, a polysaccharide used as a thickening agent in food products. Irish moss is also rich in vitamins, minerals, and antioxidants and has been traditionally used to support respiratory health and digestive function [90,91]. Sea lettuce (*Ulva* spp.), a green macroalgae that grows abundantly along coastlines worldwide, contains a moderate amount of protein and is rich in vitamins A, C, and K, as well as minerals like iron and iodine. Sea lettuce also contains chlorophyll and polyphenols, which have antioxidant and anti-inflammatory effects and may contribute to overall health and well-being [52].

Hijiki (*Sargassum fusiforme*), a brown macroalgae often used in Japanese cuisine, contains a moderate amount of protein along with dietary fiber, vitamins, and minerals such as calcium, magnesium, and iron. Hijiki also contains fucoidan, a polysaccharide with antioxidant and anti-inflammatory properties that may support immune function and cardiovascular health [62]. Dulse (*Palmaria palmata*) (Figure 5) is a red macroalga with a protein content that varies between 20% and 30% of its dry weight. While not as protein-rich as spirulina, dulse still provides a valuable source of dietary protein along with vitamins, minerals, and antioxidants. Dulse contains unique compounds such as polyphenols and polysaccharides, which have been studied for their potential anti-inflammatory and antioxidant properties [92].

Kombu (*Laminaria/Saccharina* spp.) (Figure 6) is a brown macroalgae that contains a moderate amount of protein along with various vitamins, minerals, and fiber. While not as protein-rich as spirulina, kelp provides valuable dietary protein and essential nutrients. It is also a rich source of iodine, which is essential for thyroid function and metabolism regulation [93].

Wakame (*Undaria pinnatifida*) (Figure 7) is a brown seaweed commonly used in Asian cuisine, particularly in Japanese dishes. While wakame is often praised for its delicious taste and unique texture, it also offers numerous health benefits. Wakame is low in calories and fat but rich in essential nutrients. It contains vitamins such as vitamin A, vitamin C, and various B vitamins. Wakame also provides minerals including calcium, magnesium, iodine, and iron, making it a nutrient-dense food choice [94]. While wakame is not as protein-rich as some other types of seaweed, like spirulina, it still contains a moderate amount of protein. Protein is essential for building and repairing tissues, supporting immune function, and maintaining overall health. Wakame is also a good source of dietary fiber, which is important for digestive health and regular bowel movements. Fiber can also help promote feelings of fullness and may aid in weight management by reducing calorie intake [95]. *U. pinnatifida* is particularly notable for its high iodine content. Iodine is essential for thyroid function and the production of thyroid hormones, which regulate metabolism and support growth and development. However, excessive iodine intake should be avoided, as it can lead to thyroid disorders [96]. Like other seaweeds, wakame contains antioxidants such as fucoxanthin and polyphenols. These compounds help protect cells from oxidative damage caused by free radicals and may have anti-inflammatory properties [97]. Consuming wakame may offer various health benefits, including supporting thyroid function, promoting heart health, and reducing inflammation. Some research also suggests that the compounds found in wakame may have anti-cancer properties, although more studies are needed to confirm these effects [98].

### 4.2. Extraction Methods and Processing Techniques

Compounds found in seaweed are diverse and multifaceted, contributing to both its nutritional value and functional properties. These compounds not only provide essential nutrients but also offer a range of health benefits [99]. For instance, proteins found in seaweed are vital for tissue repair, muscle growth, and immune function, while polysaccharides contribute to digestive health and may have prebiotic effects [100].

Extraction methods and processing techniques for obtaining seaweed proteins involve several stages aimed at efficiently isolating and purifying proteins from seaweed biomass. These methods often vary based on the specific properties of the seaweed species and the desired characteristics of the extracted proteins [101]. Aqueous extraction, one common method, utilizes water or saline solutions to solubilize proteins from seaweed biomass. This process involves maceration, filtration, and precipitation steps to separate proteins from other components like carbohydrates and lipids [102]. Alkaline extraction, on the other hand, employs alkaline solutions such as sodium hydroxide to disrupt cell walls and release proteins from the seaweed matrix [103]. Enzyme-assisted extraction involves the use of enzymes like proteases to break down protein structures and facilitate their extraction (see Table 2) [104].

Following extraction, processing techniques are employed to concentrate and purify seaweed proteins for various applications. Ultrafiltration, a popular method, utilizes membranes with specific pore sizes to concentrate proteins while removing unwanted impurities and contaminants [105]. Precipitation methods, such as salting out and acid precipitation, selectively precipitate proteins based on their solubility characteristics, allowing for further purification [106]. Drying techniques, including spray drying and freeze drying, remove moisture from protein concentrates to produce stable powders or flakes suitable for storage and transportation [107].

Extraction methods and processing techniques for obtaining seaweed proteins are pivotal for maximizing protein yield, purity, and functionality while minimizing energy consumption and environmental impact [108]. These methods continue to evolve with advances in technology, driving innovation in the utilization of seaweed proteins for a wide range of applications in food, feed, pharmaceuticals, and biotechnology. Moreover, the sustainable nature of seaweed cultivation underscores its potential as a valuable resource for addressing global challenges in nutrition, health, and environmental sustainability [109].

It is vital to keep in mind that the processes and methods applied are determined by the specific goals regarding protein extraction as well as the specific target proteins’ characteristics. Protein biochemistry is constantly evolving as novel methods and technologies for enhanced protein extraction, purification, and analysis emerge. Studies from the literature have extensively investigated seaweed protein concentration and purification methods to optimize yield, purity, and functionality [104]. Research has explored the effectiveness of different extraction techniques, such as aqueous extraction, alkaline extraction, and enzyme-assisted extraction, in isolating proteins from seaweed biomass [106]. These studies often evaluate factors like extraction efficiency, protein yield, and preservation of protein functionality during processing [107]. Additionally, researchers have examined various processing techniques, including ultrafiltration, precipitation methods, and drying techniques, to further concentrate and purify seaweed proteins. By assessing the impact of these methods on protein quality and functionality, researchers aim to develop efficient and sustainable approaches for harnessing seaweed proteins for diverse applications in food, feed, pharmaceuticals, and biotechnology [104].

## 5. Applications of Seaweed Proteins in Food Products, Dietary Supplements, and Biotechnological Industries

Seaweed proteins have gained increasing attention for their versatile applications in various industries, including food products, dietary supplements, and biotechnology. Here is an overview of how seaweed proteins are utilized in each of these sectors [110].

Seaweed proteins are used in a wide range of food products to enhance nutritional value, texture, and sensory attributes. They are often incorporated into plant-based and vegan food products as sustainable alternatives to animal-derived proteins [5]. Seaweed protein isolates and concentrates can be used in formulations for meat analogs, dairy alternatives, baked goods, snacks, and beverages. These proteins contribute to the protein content of food products while also providing essential amino acids and bioactive compounds [111]. Seaweed proteins can also act as emulsifiers, stabilizers, and texturizers, improving the texture and mouthfeel of food products [112].

Seaweed proteins are increasingly utilized in dietary supplements for their nutritional benefits and potential health-promoting properties. Protein powders and supplements containing seaweed proteins are marketed for athletes, fitness enthusiasts, and individuals looking to increase their protein intake. Seaweed protein supplements are also available in combination with other nutrients, vitamins, and minerals to support overall health and well-being [113]. Additionally, seaweed-based supplements may offer specific health benefits, such as supporting immune function, promoting digestive health, and enhancing skin and hair health [114].

The discussion on the assayed bioactivity of seaweed proteins and their potential implications for human health is reinforced by concrete examples showcasing their diverse range of beneficial properties [91]. For instance, studies have demonstrated the antioxidant capacity of seaweed proteins through assays measuring their ability to scavenge free radicals, such as the 2,2-diphenyl-1-picrylhydrazyl (DPPH) assay [84]. Certain seaweed proteins, like phycobiliproteins found in red algae, exhibit strong antioxidant activity, protecting cells from oxidative damage associated with aging and disease [115]. Additionally, research (see Khursheed et al., 2023 work [116]) has revealed the anti-inflammatory effects of seaweed proteins through in vitro and in vivo studies [85,116]. Furthermore, the antimicrobial and antiviral properties of seaweed proteins have been elucidated through experiments demonstrating their ability to inhibit the growth of pathogenic bacteria and viruses. For instance, lectins isolated from green seaweeds have exhibited potent antimicrobial activity against a wide range of pathogens, including antibiotic-resistant strains [106]. These concrete examples underscore the potential of seaweed proteins to positively impact human health by combating oxidative stress, inflammation, and microbial infections, thereby offering novel avenues for preventive and therapeutic interventions in various diseases [84].

Seaweed proteins are valuable resources in biotechnological industries for various applications, including pharmaceuticals, nutraceuticals, cosmetics, and renewable energy. Seaweed extracts and protein isolates are used in the development of functional foods, dietary supplements, and pharmaceutical formulations targeting specific health conditions [8]. Biotechnological processes are also employed to extract bioactive compounds from seaweed proteins for use in skincare products, wound healing formulations, and anti-aging treatments [117]. Furthermore, seaweed biomass serves as a renewable source of bioenergy through processes like anaerobic digestion and bioethanol production [118].

Seaweed proteins offer diverse opportunities for innovation and product development across the food, dietary supplement, and biotechnological industries. Their sustainable production, nutritional value, and functional properties make them attractive ingredients for meeting consumer demand for healthier and more environmentally friendly products. As research and technology continue to advance, the potential applications of seaweed proteins are expected to expand further, contributing to the development of novel products and solutions that promote health, sustainability, and innovation [119].

### 5.1. Sustainability and Environmental Considerations in Seaweed Protein Production

Sustainability and environmental considerations are paramount in seaweed protein production, reflecting the growing awareness of the importance of responsible resource management and conservation. Seaweed cultivation and processing methods are designed to minimize environmental impact while maximizing efficiency and resource utilization [120].

Sustainable seaweed protein production begins with responsible cultivation practices. Seaweed is often farmed in coastal areas using methods such as rope cultivation or integrated multitrophic aquaculture systems. These cultivation methods minimize habitat disruption and avoid the use of harmful chemicals or fertilizers that can adversely affect marine ecosystems [121]. Furthermore, seaweed cultivation can help mitigate the effects of eutrophication by absorbing excess nutrients from the water, improving water quality, and promoting biodiversity [122].

Seaweed is an inherently sustainable crop due to its rapid growth rate, high productivity, and minimal resource requirements. Unlike land-based crops, seaweed cultivation does not require arable land, freshwater, or chemical inputs such as pesticides and fertilizers. Seaweed absorbs nutrients directly from seawater and utilizes sunlight for photosynthesis, making it highly efficient in resource utilization [123]. Additionally, seaweed farming can be integrated with other aquaculture practices, such as fish or shellfish farming, to create synergies and minimize waste [122].

Seaweed plays a crucial role in carbon sequestration and climate change mitigation. Through photosynthesis, seaweed absorbs carbon dioxide from the atmosphere and converts it into biomass, effectively removing carbon from the environment. Large-scale seaweed cultivation has the potential to sequester significant amounts of carbon and offset greenhouse gas emissions, contributing to climate change mitigation efforts [11].

Sustainable seaweed protein production prioritizes biodiversity conservation and ecosystem health. Seaweed farms serve as artificial reefs, providing habitat and refuge for various marine species. By enhancing biodiversity and ecosystem resilience, seaweed cultivation helps maintain healthy marine ecosystems and supports fisheries productivity [124]. Additionally, sustainable harvesting practices ensure that seaweed populations are not depleted beyond their natural capacity to regenerate, preserving ecosystem balance and integrity [125].

Proper waste management is essential in seaweed protein production to minimize environmental pollution and nutrient runoff. Seaweed processing facilities should implement measures to reduce, reuse, and recycle waste materials, such as byproducts from protein extraction and processing. Organic waste can be composted or used as fertilizer, while non-biodegradable waste should be disposed of responsibly to prevent pollution in coastal environments [126].

Sustainability and environmental considerations are integral to the production of seaweed proteins. By adopting responsible cultivation practices, maximizing resource efficiency, promoting carbon sequestration, preserving biodiversity, and implementing effective waste management strategies, the seaweed industry can minimize its ecological footprint and contribute to the long-term health and resilience of marine ecosystems [127]. Embracing sustainability principles ensures that seaweed protein production remains environmentally sound, socially responsible, and economically viable for generations to come [6].

### 5.2. Future Perspectives

Seaweed proteins represent a promising and sustainable solution for meeting global protein demands while addressing environmental, health, and societal challenges. As evidenced by their nutritional richness, functional versatility, and eco-friendly cultivation, seaweed proteins offer a viable alternative to conventional protein sources. With their rapid growth rates, minimal resource requirements, and ability to thrive in diverse marine environments, seaweeds are poised to play a pivotal role in the sustainable food systems of the future [5].

Moreover, the multifaceted applications of seaweed proteins across food, dietary supplements, and biotechnological industries underscore their potential to contribute to diverse and innovative product offerings. From plant-based meat alternatives to functional food ingredients and beyond, seaweed proteins offer a wide range of possibilities for creating nutritious, flavorful, and environmentally friendly products that resonate with consumers worldwide [110].

Looking ahead, several key areas warrant further exploration and development to fully realize the potential of seaweed proteins. Continued research is needed to better understand the nutritional composition, functional properties, and health benefits of seaweed proteins. Innovation in extraction methods, processing technologies, and product formulation will drive the development of novel seaweed-based products with improved taste, texture, and functionality [8,105].

Efforts to enhance the sustainability and scalability of seaweed cultivation practices are essential for meeting growing protein demands without compromising marine ecosystems or coastal communities. Investment in sustainable aquaculture infrastructure, resource management strategies, and certification programs can help ensure responsible seaweed farming practices globally [6,120].

Educating consumers about the nutritional value, environmental benefits, and culinary versatility of seaweed proteins is crucial for fostering market adoption and consumer acceptance. Strategic marketing campaigns, culinary initiatives, and product innovation can help raise awareness and build demand for seaweed-based products among diverse consumer segments [128,129].

Collaboration among governments, industry stakeholders, research institutions, and civil society is essential for fostering an enabling policy environment and supporting the growth of the seaweed protein industry. Policy frameworks that incentivize sustainable practices, promote research and development, and facilitate market access can accelerate the adoption of seaweed proteins as a mainstream protein source.

Seaweed proteins have the potential to play a transformative role in meeting global protein demands while advancing sustainability, health, and innovation in the food industry. By embracing a holistic approach that prioritizes sustainability, innovation, and collaboration, seaweed proteins can emerge as a cornerstone of future food systems, contributing to a healthier, more resilient, and more sustainable world [8].

As the worldwide demand for sustainable and plant-based food choices grows, seaweed proteins can play a significant part in meeting the requirements for protein while reducing harmful effects on the environment. Seaweed proteins can be utilized to generate novel plant-based food products such as animal replacements, dairy substitutes, and baked foods. Collaborations involving scientists, seaweed farmers, food companies, and biotechnology companies can encourage innovation and speed the development of seaweed protein uses. While the potential for seaweed proteins is enormous, issues regarding flavor acceptability, production on a large scale, and cost-effectiveness must be tackled. Continuous research, investments, and cooperation throughout domains is going to be critical in achieving the full potential of seaweed proteins to shape the foreseeable future of sustainable and healthy food resources.

## 6. Conclusions

The seaweed proteins and amino acids possess multiple health benefits, including antihypertensive, antioxidant, antidiabetic, antiatherosclerosis, anti-inflammatory, antitumoral, antibacterial, antiviral, and neuroprotective properties, among many others. Humans are becoming more conscious today than ever regarding their well-being and diet, which has created a substantial market for high-value functional foods derived from natural sources rather than manufactured ingredients. However, proteins generated by seaweeds are not frequently used as ingredients in the food industry business, and few studies evaluating acceptance, harmful effects, allergy susceptibility, or microbial effects have been undertaken for seaweed.

As science and technology in this area progress, algae-based proteins can become increasingly common among alternative protein sources, contributing to the overall sustainability of the food system. Although more research and investigation need to be performed, developments in new technologies must be centered around discovering valuable compounds that promote health and removing biochemical/microbiological dangers, among other present challenges such as sustainable aquaculture and seaweed processing and preparation.

## Figures and Tables

**Figure 1 nutrients-16-01123-f001:**
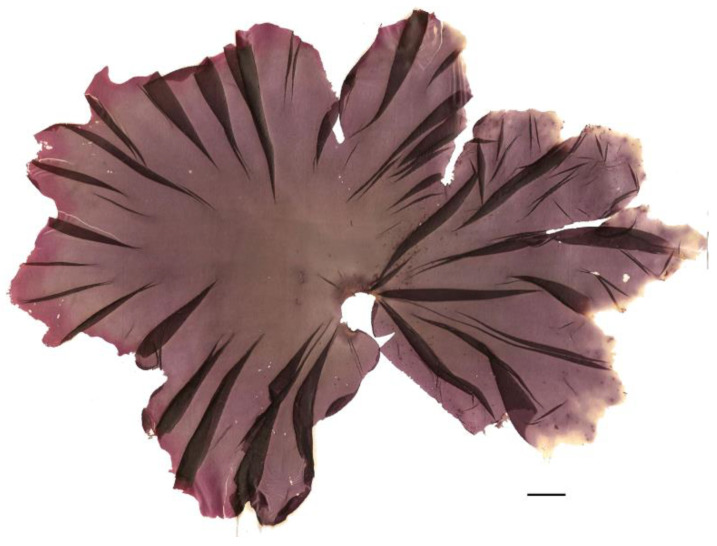
Red alga, *Porphyra umbilicalis*. Scale bar = 1 cm.

**Figure 2 nutrients-16-01123-f002:**
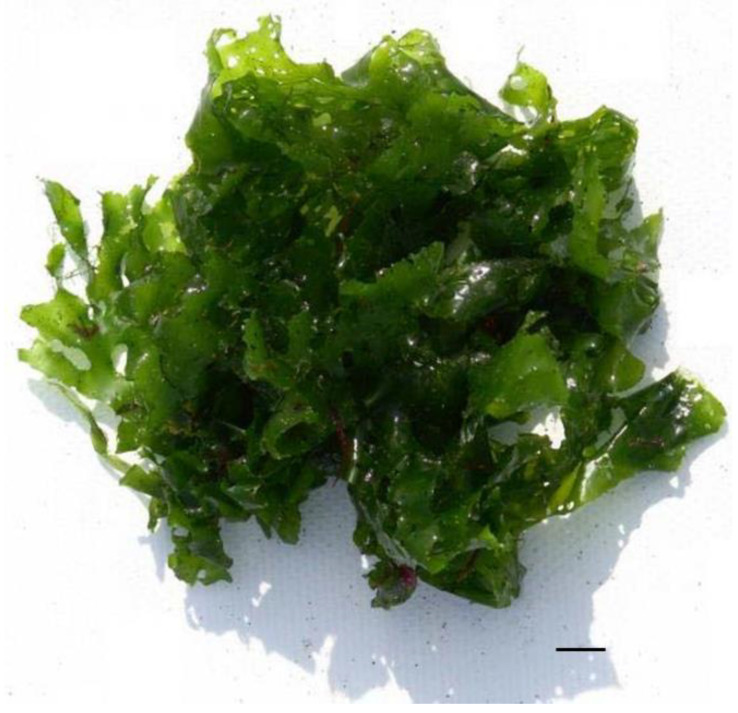
Green alga, *Ulva rigida*. Scale bar = 1 cm.

**Figure 3 nutrients-16-01123-f003:**
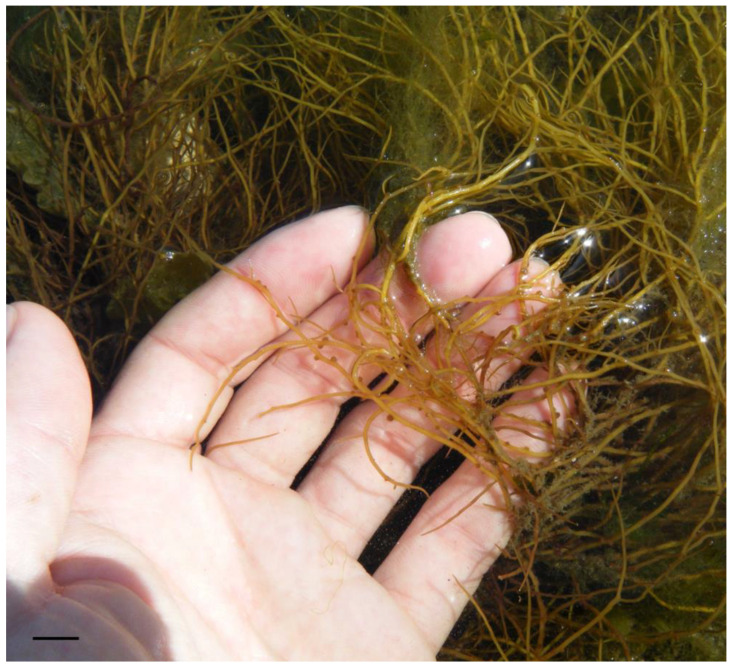
Red alga *Gracilaria gracilis*. Scale bar = 1 cm.

**Figure 4 nutrients-16-01123-f004:**
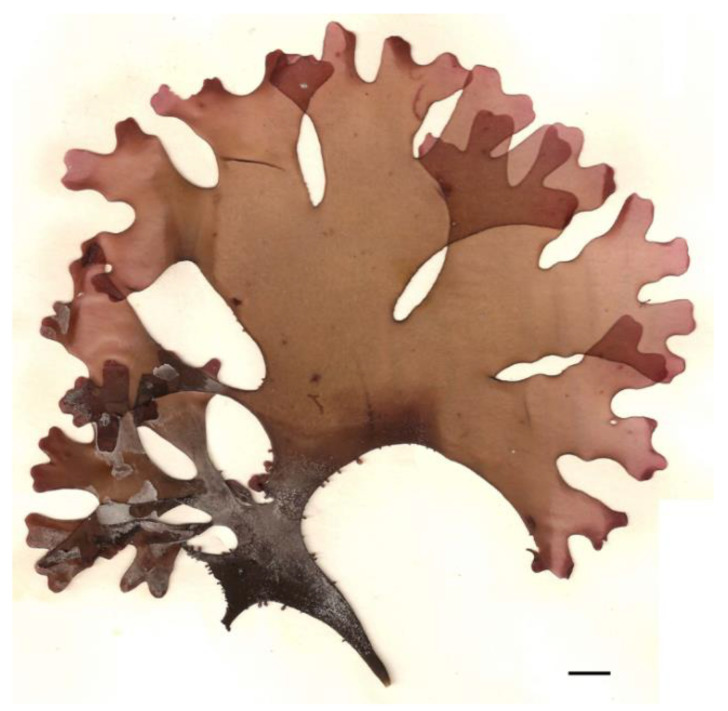
Red alga, *Chondrus crispus*. Scale bar = 1 cm.

**Figure 5 nutrients-16-01123-f005:**
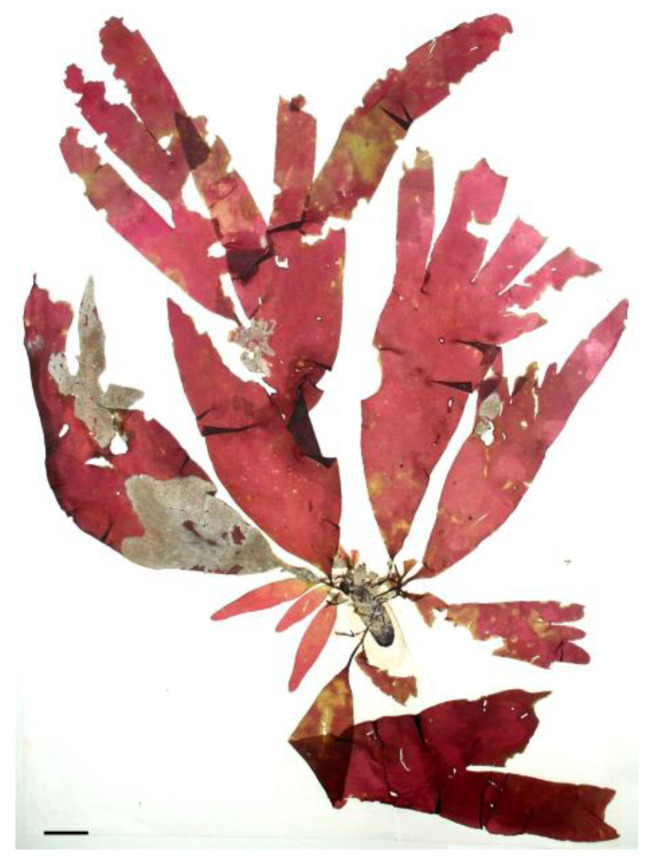
Red alga, *Palmaria palmata*. Scale bar = 1 cm.

**Figure 6 nutrients-16-01123-f006:**
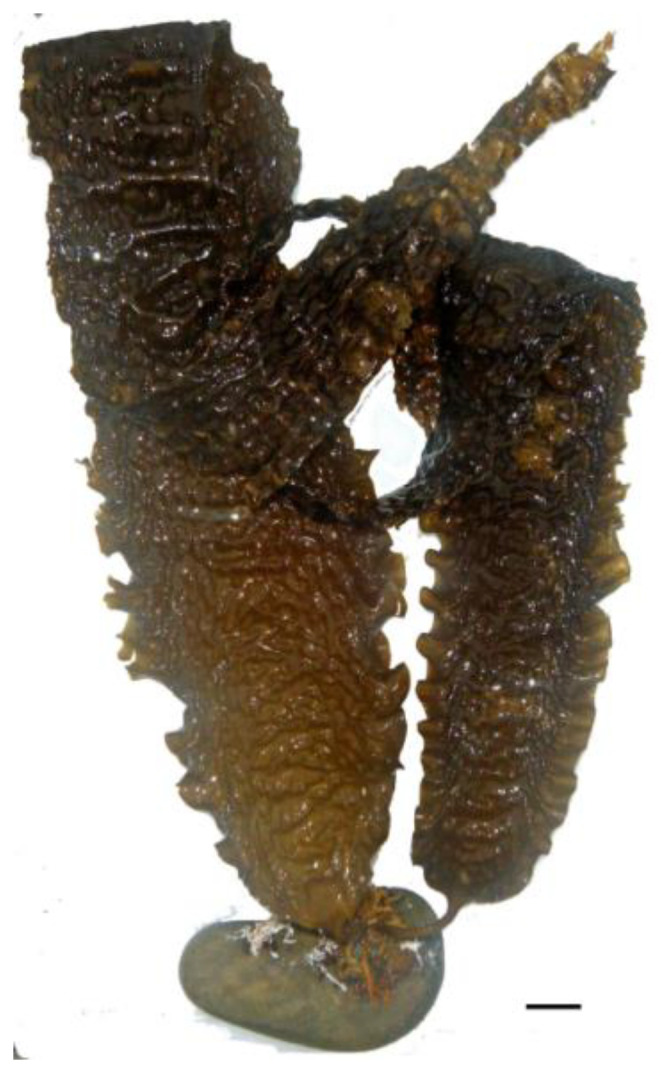
Brown alga *Saccharina latissima*. Scale bar = 1 cm.

**Figure 7 nutrients-16-01123-f007:**
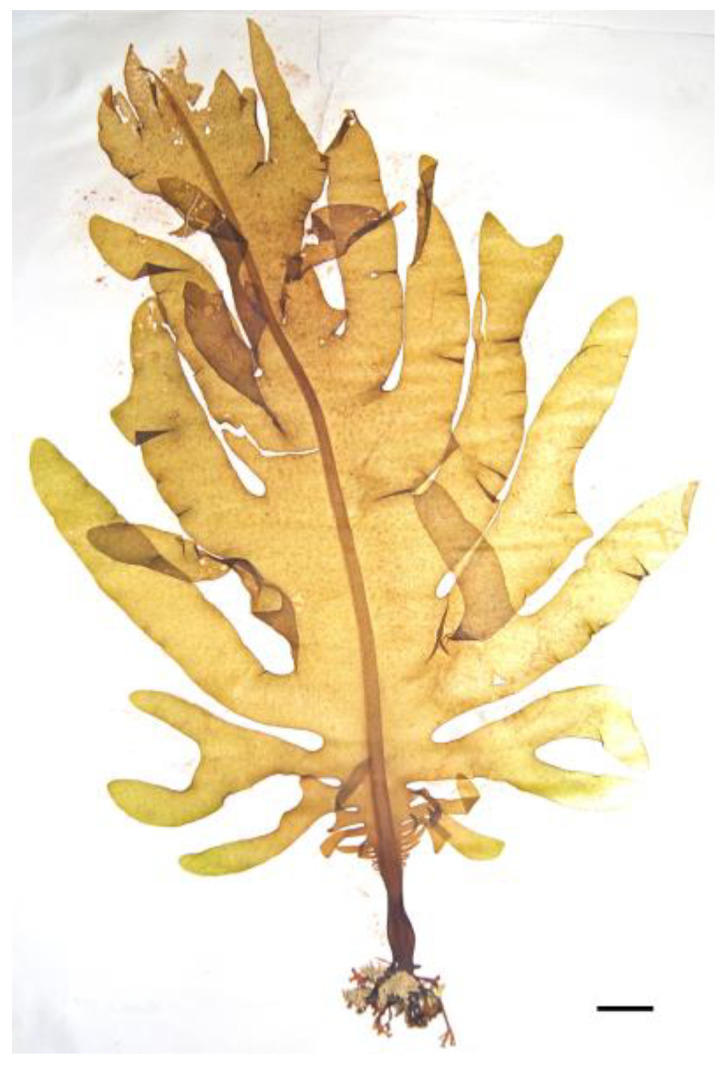
Brown alga, *Undaria pinnatifida*. Scale bar = 1 cm.

**Table 1 nutrients-16-01123-t001:** Nutrient composition of selected edible seaweed (% dry weight).

Species	Protein	Ash	Dietary Fiber	Carbohydrates	Lipid	Reference
Green seaweeds						
*Caulerpa lentillifera*	12.56–14.76	25.31–63.83	33.44–37.16	2.32–50.71	0.86–1.70	[45,46]
*Caulerpa racemosa*	17.8–19.9	7–29.4	64.9	33–41	4.5–9.8	[46,47,48]
*Codium fragile*	8–11	21–39	5.1	39–67	0-5–1.5	[48,49]
*Ulva compressa*	15.66–27	18.03–18.6	33–45	14.45–48.2	0.3–1.67	[48,49,50]
*Ulva lactuca*	10–25	11.2–12.9	28.4–38	36–58.1	0.6–1.6	[48,51]
*Ulva australis* (formerly *U. pertusa*)	14.6–26	25.9–28.6	52.2–59.0	47.0	2.1–7.4	[48,52,53]
*Ulva rigida*	15.78–19.0	20.6–28.6	38–41	17.62–56	0.9–2.0	[48,49,50]
*Ulva reticulata*	17–21.06	17.58	65.7	50–58	1.7–2.3	[48,54]
Brown seaweeds						
*Alaria esculenta*	9–20	24.2	42.86	40.7–51	1–2	[48,55]
*Eisenia bicyclis*	7.5	9.72	10–12	60.6	0.1	[48,56]
*Fucus spiralis*	9.7–10.77	-	63.88	17.6	5.23	[48,57]
*Fucus vesiculosus*	3–14	14–30	45–59	46.8	1.9–3.75	[48,58]
*Himanthalia elongata*	5–15	30–36	33–53.3	44–61	0.5–1.1	[48,59]
*Laminaria digitata*	8–15	37.59	37.3	48	1.0	[48,60]
*Saccharina japonica*	5.72–7.5	15.11–26.63	10–36	51.9–66.19	1.0–1.69	[48,61]
*Saccharina latissima*	6–26	26.2–34.78	30	52–61	0.5–1.1	[48,55]
*Sargassum fusiforme*	11.6–18.41	16.63–19.77	11.3–62	30.6–61.85	1.4–1.8	[48,62]
*Undaria pinnatifida*	12–23	26–39	16–46	45–51	1.5–4.5	[48,63]
Red seaweeds						
*Chondrus crispus*	11–21	21.08	10–34	55–68	1.0–3.0	[48]
*Gracilaria chilensis*	13.7	18.9	-	66.1	1.3	[48,64]
*Palmaria palmata*	8–35	15–30	28.57	46–56	0.7–3	[48,65]
*Pyropia tenera* (formerly *Porphyra tenera*)	33–47	9.07–20.5	12–35	44.3	0.7–2.25	[48,66]
*Porphyra umbilicalis*	29–40	10–12	29–35	43	0.2–0.3	[48,67]
*Pyropia yezoensis*	31–51.41	7.8	48.6	44.4–50.69	2.1	[48,68]

**Table 2 nutrients-16-01123-t002:** Protein extraction methods and reported protein yields (% dry weight). Adapted from O’Brien et al., 2022 [29].

Species	Extraction Methods	Protein Extracted (% Dry Weight)
*Alaria esculenta* (P)	High-pressure extraction	15%
*Alaria esculenta* (P)	Autoclave extraction	17.1%
*Ascophyllum nodosum* (P)	Chemical extraction	7.97–16.90%
*Chondracanthus chamissoi* (R)	Enzyme extraction with α-amylase, cellulose, and pectinase	36.1%
*Chondrus crispus* (R)	Osmotic shock	35.5%
*Chondrus crispus* (R)	Enzyme extraction with cellulose	7.1%
*Codium tomentosum* (C)	Ultrasound extraction and enzyme extraction with viscozyme^®^ L, cellulose (EC 232.734.4), alcalase^®^ (EC 3.4.21.14), and favourzyme^®^ (EC 232.752.2)	16.9–18.8%
*Fucus vesiculosus* (P)	Autoclave extraction	24.3%
*Fucus vesiculosus* (P)	High-pressure extraction	23.7%
*Himanthalia elongata* (P)	Chemical extraction	6.5%
*Palmaria palmata* (R)	Enzyme extraction with Umamizyme ^TM^ and xylanase	33.4%
*Palmaria palmata* (R)	Enzyme extraction with Celluclast (EC 232-734-4), Shearzyme^®^, alcalase^®^, and viscozyme^®^	35.5–41.6%
*Palmaria palmata* (R)	Autoclave	21.5%
*Porphyra umbilicalis* (R)	Chemical extraction	22.5%
*Pyropia acanthophora* (R)	Homogenisation and protein precipitation	8.9%
*Pyropia tenera* (R)	Enzyme extraction with Prolyve^®^1000, and Flavourzym^®^	23%
*Saccharina japonica* (P)	Enzyme extraction with AMG, Celluclast, Dextrozyme, Promozyme^®^, Viscozyme^®^, Alcalase^®^, Flavourzyme^®^, Neutrase, Protamex^®^, and pepsin (EC 3.4.23.1)	6.94–22.5%
*Saccharina latissima* (P)	Chemical extraction	9%
*Sargassum* spp. (P)	Enzyme extraction with cellulose and β-glucosidase	10.2%
*Sargassum wightii* (P)	Chemical extraction	8–12.2%
*Ulva* spp. (C)	Chemical extraction	8–12.2%
*Ulva lactuca* (C)	pH shift extraction with pH 2 and pH 13	22.7%
*Ulva lactuca* (C)	Enzyme extraction with papain (EC 3.4. 22.2)	69.19%
*Ulva lactuca* (C)	Enzyme extraction with endopeptidase, cellulase, xylanase, β-glucanase, arabanase	6.2–10.1%
*Ulva ohnoi* (C)	Solvent extraction	12.28–21.57%

C—Chlorophyta; P—Phaeophyceae; R—Rhodophyta.

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
