# Peer review of "Seaweed Proteins: A Step towards Sustainability?"

_nutrients, 2024, doi:10.3390/nu16081123_

Round 1
Reviewer 1 Report
Comments and Suggestions for Authors
I have carefully read the manuscript entitled “Seaweed Proteins: a step towards to sustainability?” and analyzed its potential for publication in the MDPI journal Nutrients (ISSN 2072-6643).
In my opinion, the manuscript meets the criteria of reasonable scientific article.
Below you can find my detailed remarks regarding the manuscript.
1. It would be advisable to include a clear definition of seaweed as a high-protein food derived from algae in the introduction. In the present form of the text, the reader can only guess at it. In the case of ambiguity, it is possible to misunderstand statements such as those in line 273, where the authors write: “seaweed proteins are often low in fat and calories”. During the first reading, I learned that according to the authors, proteins contain fats, which is total nonsense. Only after thoughtful consideration I realized that perhaps it was not about proteins sensus stricto, but about high-protein food. I would suggest looking at the text from this angle so as not to expose the reader to cognitive shocks.
2. Figure 4. contains structure of amino acid but it is not L-leucine as it is stated in the caption. Please correct the structure.
3. Figure 8. caption is on the other page than its graphical content, Please move the caption to the same page.
Author Response
Authors: The authors would like to thank you for your comments and suggestions for corrections.
The following paragraph was introduced in the introduction:”Seaweed as a high-protein food refers to various marine algae that contain a significant amount of protein relative to their overall composition. Seaweed is known for its rich nutritional profile, which includes vitamins, minerals, antioxidants, and other beneficial compounds [3]. While the protein content can vary depending on the type of seaweed, it generally ranges from 10% to 30% of its dry weight. Seaweed is considered a valuable source of plant-based protein, especially for individuals following vegetarian or vegan diets, as it provides essential amino acids necessary for human health. Incorporating seaweed into the diet can contribute to meeting daily protein requirements while also offering other health benefits associated with its nutrient content [4].“
About line 273, Correction: “Furthermore, seaweeds are often low in fat and calories, making them an attractive option for individuals seeking to manage their weight or reduce their intake of saturated fats [80].”
- Figure 4. contains structure of amino acid but it is not L-leucine as it is stated in the caption. Please correct the structure.
Authors: Corrected.
- Figure 8. caption is on the other page than its graphical content, Please move the caption to the same page.
Authors: Corrected.
Reviewer 2 Report
Comments and Suggestions for Authors
Comments on the manuscript “Seaweed Proteins: a step towards to sustainability?”, Manuscript ID nutrients-2940793.
The suggestions (recommendations) were made by the authors. Below, it is pointed out the specific comments:
1- Abstract: The abstract needs to be enhanced and complemented as it does not reflect the information presented in the review text. For instance, it lacks details about the types of seaweeds, extraction methods, functional properties, potential health benefits of seaweed protein, and prospects for applications.
2- In section 2. Nutritional composition of seaweed: I suggest including information on seaweed cultivation and its relationship with composition, particularly in terms of protein.
3- On page 13, section 4.2 Extraction methods and processing techniques. I suggest including a table with information from the literature, including: type of seaweed, extraction methods, operational conditions, extraction yield, and purification/concentration factor. Furthermore, in this section, studies from the literature involving protein concentration and purification methods could also be reported.
Author Response
Authors: The authors would like to thank you for your comments and suggestions for corrections.
New Abstract: “This review delves into the burgeoning field of seaweed proteins as promising alternative sources of protein. With global demand escalating and concerns over traditional protein sources' sustainability and ethics, seaweed emerges as a viable solution, offering high protein content and minimal environmental impact. Exploring the nutritional composition, extraction methods, functional properties, and potential health benefits of seaweed proteins, this review provides a comprehensive understanding. Seaweed contains essential amino acids, vitamins, minerals, and antioxidants. Its protein content ranges from 11% to 32% of dry weight, making it valuable for diverse dietary preferences, including vegetarian and vegan diets. Furthermore, the review underscores the sustainability and environmental advantages of seaweed protein production com-pared to traditional sources. Seaweed cultivation requires minimal resources, mitigating environmental issues like ocean acidification. As the review delves into specific seaweed types, ex-traction methodologies, and functional properties, it highlights the versatility of seaweed proteins in various food products, including plant-based meats, dairy alternatives, and nutritional supplements. Additionally, it discusses the potential health benefits associated with seaweed proteins, such as their unique amino acid profile and bioactive compounds. Overall, this review aims to provide insights into seaweed proteins' potential applications and their role in addressing global protein needs sustainably “
2- In section 2. Nutritional composition of seaweed: I suggest including information on seaweed cultivation and its relationship with composition, particularly in terms of protein.
Authors: These paragraphs on the suggested topic were added: “Seaweed cultivation practices significantly influence its nutritional composition, particularly concerning protein content. Cultivated seaweed tends to exhibit higher protein levels compared to wild algae [6,12]. This disparity arises due to the controlled environments of seaweed farms, where nutrient-rich conditions promote robust growth and protein accumulation. In contrast, wild algae often grow in nutrient-limited marine habitats, resulting in comparatively lower protein content [6].
Various factors during seaweed cultivation contribute to the observed differences in protein content. For instance, the availability of essential nutrients such as nitrogen and phosphorus directly impact seaweed growth and protein synthesis. Cultivation methods, such as integrated multi-trophic aquaculture (IMTA), where seaweed is grown alongside other marine organisms like shellfish, can enhance nutrient availability and thus increase protein content [13,29].
Additionally, environmental conditions, including temperature, light intensity, and salinity, play crucial roles in shaping seaweed composition. Optimal conditions promote efficient photosynthesis and metabolic processes, leading to higher protein accumulation. Moreover, the developmental stage of seaweed during cultivation can influence protein content, with certain growth phases favoring protein synthesis [6,12].
Understanding these relationships between cultivation methods and seaweed composition is essential for optimizing protein yield and nutritional quality. By elucidating the impact of cultivation practices on seaweed protein content, this review provides valu-able insights into harnessing seaweed as a sustainable and nutritious protein source for various applications [28].”
3- On page 13, section 4.2 Extraction methods and processing techniques. I suggest including a table with information from the literature, including: type of seaweed, extraction methods, operational conditions, extraction yield, and purification/concentration factor. Furthermore, in this section, studies from the literature involving protein concentration and purification methods could also be reported.
Authors: These paragraphs on the suggested topic were added: “Studies from the literature have extensively investigated seaweed protein concentration and purification methods to optimize yield, purity, and functionality [104]. Research has explored the effectiveness of different extraction techniques, such as aqueous extraction, alkaline extraction, and enzyme-assisted extraction, in isolating proteins from seaweed biomass [106]. These studies often evaluate factors like extraction efficiency, protein yield, and preservation of protein functionality during processing [107]. Additionally, researchers have examined various processing techniques, including ultrafiltration, precipitation methods, and drying techniques, to further concentrate and purify seaweed proteins. By assessing the impact of these methods on protein quality and functionality, researchers aim to develop efficient and sustainable approaches for harnessing seaweed proteins for diverse applications in food, feed, pharmaceuticals, and biotechnology [104].”
Authors: It was added a Table 2. Protein extraction methods and reported protein yields (% dry weight).
Reviewer 3 Report
Comments and Suggestions for Authors
The manuscript entitled "Seaweed Proteins: a step towards to sustainability?" seeks to explore the emerging field of seaweed proteins as uncommon yet promising alternative sources of protein. The review article is well structured and informative however minor editing of english language is needed.
Comments on the Quality of English LanguageMinor editing of english language needed
Author Response
Authors: The authors would like to thank you for your comments and suggestions for corrections. The manuscript was revised for English.